The transition to agricultural cultivation of neo-crops may fail to account for wild genetic diversity patterns: insights from the Cape Floristic Region

Galuszynski Nicholas C. nicholas.galuszynski@gmail.com
Department of Botany, Nelson Mandela University , Gqebehra , South Africa
Sosa Victoria
Electronic publication date: 2021 Jun 9
Publication date: 2021
Volume: 9
Electronic Location ID: e11462
Received 2020 Dec 17; Accepted 2021 Apr 26
Copyright: ©2021 Galuszynski
Copyright year: 2021
Copyright holder: Galuszynski
License: This is an open access article distributed under the terms of the Creative Commons Attribution License, which permits unrestricted use, distribution, reproduction and adaptation in any medium and for any purpose provided that it is properly attributed. For attribution, the original author(s), title, publication source (PeerJ) and either DOI or URL of the article must be cited.
License URL: https://creativecommons.org/licenses/by/4.0/

Keywords: Genetic Diversity, Honeybush, Wild Genetic Resources, Conservation Genetics, Genetic Risk, Genetic Pollution, Applied Phylogeography, Phylogeography

Funding: National Research Fund of South Africa 99034 95992 114687 Table Mountain Fund TM2499 This work was supported by the National Research Fund of South Africa (Grant nos. 99034, 95992, 114687) and the Table Mountain Fund (Grant no. TM2499). The funders had no role in study design, data collection and analysis, decision to publish, or preparation of the manuscript.

==============================
Aim

The global increase in the cultivation of native wild plants has raised concerns regarding potential risks associated with translocating genetic lineages beyond their natural range. This study aimed to investigate whether agricultural cultivation of neo-crops (a) accounts for the levels of genetic diversity present in wild populations, and whether (b) cultivated populations are genetically divergent from wild populations and thus pose a potential threat to wild genetic diversity.

Location

The Cape Floristic Region (CFR), located along the southern Cape of South Africa.

Methods

High Resolution Melt analysis (HRM) coupled with Sanger sequencing was used to screen three non-coding chloroplast DNA loci in Cyclopia Vent. (Fabaceae), a CFR endemic neo-crop cultivated for the production of a herbal infusion referred to as Honeybush tea. Wild and cultivated populations for three of three widely cultivated Honeybush species (C. intermedia, C. longifolia, and C. subternata) were screened. Genetic diversity and differentiation were measured and compared between wild and cultivated groups.

Results

Across all asseccions, a total of 17 haplotypes were detected, four of which were shared between wild and cultivated populations, while the remaining 13 were only detected in wild populations. Genetic diversity and differentiation was significantly higher in wild populations than in cultivated populations.

Conclusions

If no guidelines exist to facilitate the introduction of native wild plant taxa to a cultivated setting, wild genetic diversity patterns are likely to be compromised by cultivated populations. In the case presented here, cultivation represents a genetic bottleneck, failing to account for rare haplotypes, and may have disrupted species boundaries by initiating interspecific hybridization. More empirical work is required to evaluate the extent to which neo-crop cultivation poses a risk to wild genetic resources in the CFR and globally.

Introduction

Commercial trade of wild crop and medicinal plants relies predominantly on material sourced from wild populations. However, consumer demand for ’natural products’ and products manufactured from renewable resources has promoted an increase in the domestication and cultivation of wild crop species (Lubbe & Verpoorte, 2011), referred to here as neo-crops. If managed effectively, cultivation could facilitate the preservation of these economically important species by safeguarding genetic diversity ex-situ. Alternatively, widespread cultivation may reduce incentives to protect natural populations, with natural populations being replaced by cultivated ones. Additionally, the cultivation of commercial strains adjacent to natural populations may increase the risk of exposing wild plants to non-local genetic lineages that may result in hybridization, and thus, genetic erosion (Hammer & Teklu, 2008; Laikre et al., 2010).

Gene flow from cultivated to wild populations is particularly common in traditional crop systems (Ellstrand, Prentice & Hancock, 1999) and can negatively impact wild populations by disrupting local genetic diversity and adaptation (Laikre et al., 2010). Neutral and selective pressures result in the divergence of ecological and genetic traits among geographically separate populations (Hufford & Mazer, 2003). When this geographic separation is overcome due to anthropocentric translocation of genetic material, the fate of non-local genes that escape into wild populations is challenging to predict. There have, however, been cases where non-local lineages have invaded native populations, resulting in a loss of local genetic diversity (e.g., Carex caryophyllea Latourr (Cyperaceae), Whitlock, Grime & Burke, 2010; and Phragmites australis (Cav.) Steud, Chambers, Meyerson & Saltonstall, 1999). A precautionary approach that limits the distance that genetic material is translocated should, therefore, be adopted during anthropogenic redistribution of genetic material (Byrne & Stone, 2011; Galuszynski & Potts, 2020a). This is rarely the case, and the evolutionary history of the taxa involved is often not considered. Rather, in many cases, seed is sourced from distant populations or seed lots and have already undergone some form of screening for individuals with commercially favorable traits (Hyten et al., 2006; Schipmann et al., 2005; Tembrock et al., 2017; Yuan et al., 2010), that may be detrimental when introduced to natural populations.

Cultivated populations are likely to be poor representatives of local genetic diversity. In the case of the stimulant plant qat (Catha edulis [Vahl] Forssk. ex Endl., Celastraceae), genotypes sourced from wild populations in Ethiopia were used to establish cultivated populations in Yemen and Kenya, yet both regions support genetically distinct natural qat populations that could have formed the basis for local cultivation (Tembrock et al., 2017). In contrast, cultivated populations of the Chinese skullcap (Scutellaria baicalensis Georgi, Lamiaceae) contained a combination of haplotypes sourced from multiple, geographically separate, wild populations, lacking the phylogeographic structuring present in wild populations (Yuan et al., 2010). Despite representing different approaches to collecting genetic material for cultivation, if gene flow were to occur from cultivated to wild populations in either of these cases, the genetic integrity of wild populations would be compromised.

The Cape Floristic Region (CFR; Goldblatt, 1978), located on the southern coast of South Africa is well known for its species richness, supporting over 9000 species in an area of approximately 90 000 km2 (Goldblatt & Manning, 2002), and home to various economically important plant species (Reinten, Coetzee & Van Wyk, 2011; Scott & Hewett, 2008; Turpie, Heydenrych & Lamberth, 2003). The high floristic diversity of this region has resulted from low extinction rates in a topographically and edaphically heterogeneous landscape, which produces steep ecological gradients and isolate populations over relatively short distances (Barraclough, 2006; Cowling, Ş & Partridge, 2009; Cowling et al., 2017). Genetic divergence (within and among species) in the CFR is therefore possible over relatively short distances, producing plant populations that exhibit spatially structured genetic diversity (phylogeographic structuring) (Galuszynski & Potts, 2020b; Tolley et al., 2014). The CFR is thus an ideal system for testing the representation of wild genetic variation in cultivated neo-crop plants.

The commercial trade in South African plant products relies predominantly on raw material sourced from wild populations (Van Wyk & Prinsloo, 2018). However, species used for the production of products with high export value are becoming widely cultivated (Reinten, Coetzee & Van Wyk, 2011; Turpie, Heydenrych & Lamberth, 2003). This transition to cultivation may pose a threat to the genetic integrity of wild populations of the target species, as the underlying levels and distribution of genetic diversity are not considered during the selection and translocation of commercially important CFR plants (Van Wyk, 2008). The consequences of this have already led to interspecific hybridization among Protea L. species (Proteaceae) (Macqueen & Potts, 2018) and possible genetic erosion of wild Rooibos (Aspalathus linearis (Burm.f.) R.Dahlgren, Fabaceae) (Malgas et al., 2010). As a result, concerns regarding potential genetic risk associated with a shift to widespread cultivation of Honeybush tea—a herbal infusion produced from members of the CFR endemic genus Cyclopia Vent (Fabaceae)—have been raised (Potts, 2017).

The global increase in demand for Honeybush tea, coupled with local declines in wild populations, prompted interest in cultivation (Joubert et al., 2011). Initial breeding trials included 12 of the 23 Cyclopia species and involved the selection of individuals with commercially favorable traits (e.g., bushy and vigorous growth forms), and cross pollination experiments (Joubert et al., 2011). Seed produced from these trials formed the foundation of early Honeybush cultivation, which was promoted in areas that supported natural Cyclopia populations (Jacobs, 2008; Joubert et al., 2011). This history of domestication trials and translocation is not unique to Honeybush, with a similar approach applied to neo-crops elsewhere in Southern Africa (Akinnifesi et al., 2006), Australia (Ahmed & Johnson, 2000), and South Korea (Pemberton & Lee, 1996), and may be typical of neo-crop development.

This study is the first to describe the levels of genetic diversity among wild and cultivated populations of an endemic crop plant originating from the CFR, focusing specifically on three widely cultivated Cyclopia species. High Resolution Melt analysis (HRM, Wittwer et al., 2003) coupled with sequence confirmation is applied to screen variation across two non-coding chloroplast DNA (cpDNA) regions (the atpI - atpH intergenic spacer and ndhA intron). This study explores the prediction that cultivated Cyclopia populations will fail to reflect the spatial distribution and diversity of chloroplast lineages present in the wild. Thus, the current state of honeybush cultivation, and likely neo-crop cultivation in general, may represent a genetic risk to the integrity of wild populations occurring in close proximity to cultivated populations.

Methods and Materials

Target taxa and sampling

The species selected for evaluation (C. intermedia E.Mey, C. subternata Vogel., and C. longifolia Vogel.) represent the most widely cultivated Honeybush taxa, cultivated in the Western Cape and Eastern Cape provinces of South Africa (Joubert et al., 2011; McGregor, 2017). Consequently, these three species have likely experienced the greatest extent of redistribution outside of their natural range through cultivation (Fig. 1; Fig. 3 in Joubert et al., 2011)—placing them at high risk of genetic pollution. Additionally, these species represent three distinct distribution and life history patterns (Schutte, 1997): a widespread obligate resprouter occurring at altitudes between 500–1700 m (C. intermedia), a widespread coastal lowland obligate seeder (C. subternata ) and a critically endangered Eastern Cape endemic, riparian specialist with a mixed post fire response of facultative seeding and sprouting (C. longifolia). Since life history traits and range size impact a species’ demographic history (Ellegren & Galtier, 2016), these three taxa are likely to exhibit different genetic diversity patterns that need to be accounted for during translocation and cultivation.

Figure 1 Distribution of Cyclopia populations screened for haplotype diversity.

Cyclopia populations indicated by circles, with the colors for wild and cultivated populations for each species indicated. Red stars indicate the locations of Honeybush nurseries, white pentagons indicate the locations of populations initially used for cultivar development for each species respectively. The natural distribution of the three target species is indicated using the same colors used to indicate wild populations and shaded. (A) Distribution of C. intermedia, in set indicates the study domain in relation to Africa and South Africa; (B) distribution of C. subternata; and (C) distribution of C. longifolia, the initial source of C. longifolia breeding material is the same location as the Longmore populations (LMF, LMR) and cultivated material was sourced from G, P, and U (in A and B and indicated by a light blue outline). Population naming follows the descriptions in Table 1. Cultivated populations: G = George, U = Uniondale, H = Harkerville, HAR = Harlem, P = Plettenberg Bay. Wild populations: GAR = Garcia’s Pass, SWB = Swartberg Mountains, LK = Langkloof, LS = Lady Slipper, OP = Outeniqua Pass, BKB = Bloukrans Bridge, KAR = Kareedouw Pass, LMF = Longmore Forest, LMR = Longmore River, SR = Sand River, VS = Van Stadens River.

Figure 2 The relationships among haplotypes from the merging chloroplast DNA regions screened via HRM, as inferred from the statistical parsimony algorithm.

Black circles indicate “missing” haplotypes, whist haplotypes connected by a single line differ by a single nucleotide mutation. Areas of circles and numerical labels correspond to the haplotype frequency. Circle colour indicates the species and source of each haplotype (as denoted by the figure key, with the size of each colour segment corresponding to haplotype frequency. Note that haplotypes A and B occur at low frequencies in some species: A occurs once in cultivated and wild C. subternata populations, and once in wild and twice in cultivated C. intermedia populations, B occurs twice in cultivated C. subternata populations. Haplotypes marked with * indicate possible cases of chloroplast capture, as these haplotypes were detected only in cultivated C. Subternata populations and wild and cultivated C. intermedia populations, but not in any wild C. subternata populations. Haplotypes frequencies for each population are given in Table 1 and nucleotide differences among haplotypes are summarized in Table 2.

Figure 3 Unrooted Neighbour Joining clustering diagram of Cyclopia populations based on pairwise population genetic distance.

Branches with over 50% bootstrap support are labeled. A scale bar of pairwise population genetic distance is provided above the diagram. Branch tips are labeled by species followed by an abbreviated population name following the descriptions Table 1. Open circles indicate wild populations while closed circles indicate cultivated populations. Wild and cultivated populations that group together are indicated by bold type face.

Two of these species (C. Intermedia and C. Subternata) were subject of recent phylogeographic studies (Galuszynski & Potts, 2020a; Galuszynski & Potts, 2020c) and these previously published data sets were bolstered with additional samples to ensure that 24 individuals were available from each populations for analysis. Samples were collected from four geographically separated wild populations across the natural range of each species. The intention was to (a) maximise the genetic variation detected among populations, and (b) provide a representative reference of haplotypes to describe the origins of material under cultivation. Cultivated material was sampled from Honeybush farms identified remotely from internet searches rather than relying on existing farmer networks, and were located in different mountain ranges (details of the locations are provided in Fig. 1 and Table 1). This approach was employed to avoid potentially redundant sampling of cultivated material originating from seed exchange between farmers located in close proximity to one another, however, the true origin of the commercial seed remains uncertain (see discussion). The cultivators included in the study are situated near to three of the four major Honeybush nurseries reported by Joubert et al. (2011) (Fig. 1). Three cultivated populations of each species were sampled. From all populations (wild and cultivated), a total of 24 plants were sampled with a minimum of 5 m distance between sampled individuals. The final data set consisted of 504 samples collected across 21 (12 wild, 9 cultivated) populations of three commercially important Cyclopia species, population locations are mapped in Fig. 1. Fresh leaf material was collected from a healthy growing tip of each individual and placed into silica desiccating medium for a minimum of two weeks prior to DNA extraction. All sampling was approved by the relevant landowners involved and permitting agencies, Cape Nature (Permit number: CN35-28-4367), the Eastern Cape Department of Economic Development, Environmental Affairs and Tourism (Permit numbers: CRO 84/ 16CR, CRO 85/ 16CR), and the Eastern Cape Parks and Tourism Agency (Permit number: RA_0185).

DNA extraction and haplotype detection

The DNA extraction and haplotype detection protocol followed the approach previously described in Galuszynski & Potts (2020c) and a brief overview of the approach is provided here. Whole genomic DNA was extracted using a modified CTAB DNA extraction approach, adapted from Doyle & Doyle (1987). Extracted DNA was quantified using a NanoDrop 2000c Spectrophotometer (Thermo Fisher Scientific, Wilmington, DE19810r Scientific, USA) and diluted to 5 ng/µL for PCR amplification and subsequent HRM analysis.

High Resolution Melt analysis involves the gradual heating of PCR products amplified in the presence of a DNA saturating dye. As the double stranded DNA is heated it dissociates at a rate based on the binding strength of the nucleotide sequence under analysis. As such, different nucleotide sequences should produce a distinct melt curve when plotting sample differences in measured fluorescence against change in temperature.

Three DNA fragments from two non-coding cpDNA regions (atpI-atpH intergenic spacer and ndhA intron) were amplified using Cyclopia specific primers and subsequently screened for nucleotide variation via HRM curve analysis. Samples were run in duplicates and HRM clustering was conducted on a single population basis following the recommendations of Dang et al. (2012). This was achieved by grouping populations using the ‘well group’ option in the CFX Manager Software (Bio-Rad Laboratories, Hercules, California, U.S.A.) and running the HRM clustering on these predefined population well groups. All reactions (PCR amplification and subsequent HRM) took place in a 96 well plate CFX Connect (Bio-Rad Laboratories, Hercules, California, U.S.A.). Haplotype melt curve grouping was achieved using the automated clustering algorithm of the High Precision Melt software (Bio-Rad Laboratories, Hercules, California, U.S.A.) (ΔTm =0.05, curve shape sensitivity = 70%, temperature correction = 20). HRM cluster to haplotype confirmation was achieved by unidirectional sequencing, as described in Galuszynski & Potts (2020c). The chloroplast regions targeted by HRM were sequenced for a subset of individuals per HRM cluster from populations not previously studied by Galuszynski & Potts (2020a); Galuszynski & Potts (2020c)). A total of 39 individuals for the atpI-atpH intergenic spacer and 36 individuals for the ndhA intron were PCR amplified using the reverse primers and following PCR protocols of (Shaw et al., 2007) and sequenced. The PCR and HRM conditions and details of the primers used in this study, are provided in S1.

Table 1 Summary of wild and cultivated Honeybush (Cyclopia) populations.

Population localities, including population name and abbreviation (used in), mountain range each population was sampled from, geographic coordinates, number of accessions screened per population (N), number of haplotypes detected per population (H) with haplotypes unique to the population given in parenthesis, and summary of haplotypes found in each population.

Origin	Species	Location	GPS co-ordinates	N	H	Haplotype	
		Population	Mountain	X	Y			A	B	C	D	E	F	G	H	I	J	K	L	M	N	O	P	Q	
Cultivated	C. intermedia	George (G)	Outeniqua	−33.93	22.32	24	2 (1)	–	23	–	1	–	–	–	–	–	–	–	–	–	–	–	–	–	
		Uniondale (U)	Kammanassie	−33.66	23.14	24	1	–	24	–	–	–	–	–	–	–	–	–	–	–	–	–	–	–	
		Harlem (HAR)	Tsitsikamma	−33.74	23.34	24	2	2	22	–	–	–	–	–	–	–	–	–	–	–	–	–	–	–	
	C. longifolia	George (G)	Outeniqua	−33.99	22.36	24	1	24	–	–	–	–	–	–	–	–	–	–	–	–	–	–	–	–	
		Uniondale (U)	Kammanassie	−33.66	23.14	23	1	23	–	–	–	–	–	–	–	–	–	–	–	–	–	–	–	–	
		Plettenberg Bay (P)	Tsitsikamma	−33.93	23.48	24	1	24	–	–	–	–	–	–	–	–	–	–	–	–	–	–	–	–	
	C. subternata	Uniondale (U)	Kammanassie	−33.66	23.14	24	3	–	2	21	–	–	–	–	–	1	–	–	–	–	–	–	–	–	
		Harkerville (H)	Outeniqua	−34.04	23.23	24	3	1	–	22	–	–	–	–	–	1	–	–	–	–	–	–	–	–	
		Plettenberg Bay (P)	Tsitsikamma	−33.93	23.48	24	1	–	–	24	–	–	–	–	–	–	–	–	–	–	–	–	–	–	
Wild	C. intermedia	Garcia’s Pass (GAR)	Langeberg	−33.96	21.22	23	1	–	–	–	–	23	–	–	–	–	–	–	–	–	–	–	–	–	
		Swartberg Mountains (SWB)	Swartberg	−33.33	22.04	23	6 (2)	1	3	–	–	–	14	3	1	1	–	–	–	–	–	–	–	–	
		Langekloof (LK)	Kouga	−33.78	23.79	24	2 (1)	–	23	–	–	–	–	–	–	–	1	–	–	–	–	–	–	–	
		Ladyslpper (LS)	Cockscomb	−33.9	25.25	24	3 (2)	–	–	–	–	–	2	–	–	–	–	21	1	–	–	–	–	–	
	C. longifolia	Longmore Forest (LMF)	Van Stadens	−33.84	25.09	23	1	23	–	–	–	–	–	–	–	–	–	–	–	–	–	–	–	–	
		Sand River (SR)	Van Stadens	−33.73	25.09	24	2 (1)	19	–	–	–	–	–	–	–	–	–	–	–	5	–	–	–	–	
		Longemore River (LMR)	Van Stadens	−33.81	25.15	23	2 (1)	19	–	–	–	–	–	–	–	–	–	–	–	–	4	–	–	–	
		Van Stadens River (VS)	Van Stadens	−33.9	25.21	24	1	24	–	–	–	–	–	–	–	–	–	–	–	–	–	–	–	–	
	C. subternata	Garcia’s Pass (GAR)	Langeberg	−33.96	21.22	24	1	–	–	–	–	24	–	–	–	–	–	–	–	–	–	–	–	–	
		Outeniqua Pass (OP)	Outeniqua	−33.88	22.4	24	3 (1)	1	–	18	–	–	–	–	–	–	–	–	–	–	–	5	–	–	
		Bloukranz Bridge (BKB)	Tsitsikamma	−33.97	23.65	22	2 (1)	–	–	17	–	–	–	–	–	–	–	–	–	–	–	–	5	–	
		Kareedou Pass (KAR)	Tsitsikamma	−33.97	24.22	23	2	–	–	19	–	–	–	–	–	–	–	–	–	–	–	–	–	4	

Table 2 Summary of chloroplast DNA nucleotide differences for the three loci screened by HRM.

Haplotype frequency in each population is reported in Table 1.

	MLT S1 –MLT S2 atpI-atpH intergenic spacer		MLT S3 –MLT S4 atpI-atpH intergenic spacer		MLT U1 –MLT U2 ndhA intron	
Position	4–10	47–110	144–150		213	230	294–301	308	314	380	388	421	443		503	597	682	717	731	799	
Consensus	1	2a	3		G	C	4	G	T	T	G	T	C		G	T	G	C	G	C	
Haplotype																					
A	1	.	.		.	.	4	.	.	.	.	c	.		.	.	.	.	.	.	
B	1	.	.		.	.	4	.	.	.	.	.	.		.	.	.	.	.	.	
C	1	.	.		.	.	4	.	.	.	a	.	.		.	.	.	.	.	.	
D	1	.	.		.	.	4	.	.	g	.	.	.		.	.	.	.	.	.	
E	1	2b	.		.	.	4	.	.	.	a	.	.		.	.	.	.	.	.	
F	1	.	.		.	.	4	.	.	.	.	.	.		.	.	t	.	t	.	
G	1	.	.		.	.	4	.	.	.	.	.	.		a	.	t	.	.	.	
H	1	.	.		.	t	4	.	.	.	.	.	.		a	.	t	.	.	.	
I	—	.	.		.	.	4	.	.	.	.	.	.		.	.	.	.	.	.	
J	1	.	—		.	.	4	a	.	.	.	.	.		.	.	.	.	.	.	
K	1	—	.		.	.	4	.	.	.	.	.	.		.	.	.	a	.	a	
L	1	2c	.		.	.	4	.	c	.	.	.	.		.	.	.	.	.	.	
M	1	.	.		.	.	4	.	.	.	.	c	t		.	.	.	.	.	.	
N	1	.	.		.	.	4	.	.	.	.	c	.		.	a	.	.	.	.	
O	1	.	.		a	.	–	.	.	.	a	.	.		.	.	.	.	.	.	
P	1	2d	.		.	.	4	.	.	.	a	.	.		.	.	.	.	.	.	
Q	1	2e	.		.	.	4	.	.	.	a	.	.		.	.	.	.	.	.	
Notes.

1 tatctaa

3 aaaattt

4 tatcccc

2a tacagatgaaaggaagggcttcgttttttgaatcctatctaaatttacagtaacagggcaaa

2b tacagatgaaaggaagggcttcgttttttgaaaactatctaaatttacagtaacagggcaaa

2c tacagatgaaaggaaggggttcgttttttgaatcctatctaaatttacagtaacagggcaaa

2d taaagatgaaaggaagggcttcgttttttgaatcctatctaaatttacagtaacagggcaaa

2e tatagatgaaaggaagggcttcgttttttgaatcctatctaaatttacagtaacagggcaaa

Sequences were assembled using CondonCode Aligner [v2.0.1] (CodonCode Corp, http://www.codoncode.com). Each base-call was assigned a quality score using the PHRED base-calling program (Ewing et al., 1998). Sequences were then automatically aligned using ClustalW (Thompson, Higgins & Gibson, 1994) and visually inspected. All indels that were difficult to score (due to homopolymer repeats that are prone to alignment errors) were removed. The cpDNA regions under investigation are maternally inherited in tandem and not subject to recombination (Reboud & Zeyl, 1994), and were therefore combined for subsequent analysis. A custom R script (provided with a minimum working example online: https://doi.org/10.6084/m9.figshare.12624620.v1) was then used to assign each sample its respective haplotype identity based on HRM clustering.

Haplotype diversity analysis

All analyses were performed in R (V 3.5.1) (R Core Team, 2018). The genealogical relationships among haplotypes were established using a Statistical Parsimony (SP) network (Fig. 2) constructed in TCS (v 1.2.1) (Clement, Posada & Crandall, 2000). As TCS treats each base pair in an indel as an evolutionary event, all indels were reduced to a single base pair prior to analysis with default options selected for network construction. Haplotype diversity and differentiation was compared between wild and cultivated individuals grouped by species and origin. Differences in gene diversity (GD) between wild and cultivated populations were tested via Mantel tests (Mantel, 1967); using Hs.test function implemented in the adegenet (v 2.1.1) library, (Jombart & Ahmed, 2011). Genetic differentiation between wild and cultivated populations were tested via an Analysis of Molecular Variance (AMOVA) (Excoffier, Smouse & Quattro, 1992) using the poppr.amova function with 999 permutations implemented in the poppr [v2.8.3] library (Kamvar, Tabima & Grünwald, 2014). Three additional population differentiation measures were calculated: two fixation indices, pairwise Gst (Nei, 1973) and G””st (Hedrick, 2005), and a measure of genetic divergence, Jost’s D (Jost, 2008), using the pairwise_Gst_Nei, pairwise_Gst_Hendrick , and pairwise_D functions respectively, all from the mmod [v1.3.3] library (Winter, 2012). Population clustering was inferred from a Neighbor Joining tree constructed using Prevostis pairwise population genetic distance (Prevosti, Ocaña & Alonso, 1975), calculated using the prevosti.dist function in the poppr library). This distance measure treats alignment gaps as evolutionary events and all gaps were reduced to a single base pair prior to analysis. Support for population clustering was assessed via a bootstrap analysis with 9999 replicates, implemented using the aboot function implemented in poppr.

Results

Haplotype detection

Of the 504 samples screened for haplotype variation, seven (six wild and one cultivated) failed to PCR amplify despite repeated efforts and the final dataset consisted of 497 samples. High Resolution Melt analysis with haplotype confirmation by sequencing revealed 17 cpDNA haplotypes with 100% specificity for all three loci (i.e., no cases of different haplotypes being grouped into the same HRM cluster were detected). The final concatenated dataset consisted of 794 bp (457 bp from the atpI-atpH intergenic spacer and 339 bp from the ndhA intron) with an overall GC content of 28.1%. The alignment contained 22 polymorphic sites including nine transitions, ten transversions, and three indels (two of 7 bp and one of 71 bp). Haplotype frequency within populations and nucleotide variation among haplotypes are summarized in Tables 1 and 2, respectively.

Genetic diversity analysis

The SP network (Fig. 2) revealed relatively low divergence among haplotypes, with all haplotypes diverging from a central variant. Of the 17 haplotypes detected, only four were present in cultivated populations. These four haplotypes (in addition to a fifth haplotype, detected only in wild C. subternata and C. intermedia populations from Garcia’s Pass, GAR Fig. 1) were shared among species. Two of these haplotypes were detected in C. intermedia populations (wild and cultivated) and cultivated C. subternata populations, but were not present in any wild C. subternata populations screened.

Clustering of Cyclopia populations, based on pairwise population genetic distance resulted in weak grouping of species. Cultivated populations, however, exhibited little differentiation and generally clustered together based on species (Fig. 3). Similarly, wild C. longifolia populations exhibited little genetic differentiation and all wild and cultivated populations of this species formed a single group (Fig. 3). Wild C. intermedia and C. subternata populations tended to exhibit higher levels of genetic divergence. All cultivated C. subternata populations were clustered with two wild populations originating from the Tsitsikamma mountains (Kareedouw Pass KAR, and Bloukrans Bridge BKB). The remaining wild C. subternata populations were more divergent and did not cluster with other C. subternata populations. The C. intermedia and C. subternata populations sampled from Garcia’s pass (GAR) exhibited no genetic differentiation (both fixed for haplotype F). In general, wild C. intermedia populations tended to be genetically distinct, supporting unique haplotypes and formed no clear clusters in the Neighbor Joining population tree (Fig. 3). Cultivated C. intermedia populations did not exhibit this variability. Rather, all cultivated C. intermedia populations form a single cluster with the wild C. intermedia population sampled from the Langkloof (LK).

In the cases of C. subternata and C. intermedia, genetic structuring was detected in wild populations, with 60.1% and 83.3% of genetic variation detected among populations for the two species, respectively (associated Fst values significant, p < 0.005, Fst =0.093 and 0.023, respectively). In contrast, no structuring was detected in cultivated populations of these species, with 4.3% and 1.7% of variation structured within cultivated populations for C. intermedia and C. subternata, respectively (p < 0.005, Fst =0.0001 for both species). No genetic structuring was found for wild or cultivated C. longifolia populations, as all populations shared the same common haplotype and only two rare haplotypes (N and M, Fig. 2) were detected in wild populations. Gene diversity and genetic differentiation followed a similar pattern, with wild C. intermedia and C. subternata populations having higher mean diversity than cultivated populations (p < 0.01), but no differences in gene diversity was detected between wild and cultivated C. longifolia populations. Mean genetic differentiation (Gst, G”st and Josts D) was higher in all wild populations than in cultivated populations. All population differentiation and diversity measures are summarized in Table 3.

Table 3 Genetic diversity, fixation and differentiation measures for wild (W) and cultivated (C) Honeybush (Cyclopia) populations.

Significance values are indicated for comparisons of mean genetic diversity, fixation and differentiation between wild and cultivated populations for each species and all species pooled (Total).

Species	Source	N	Hr	GD	Gst (SD)	G”st	Jost’s D	Genetic variation (%)	
C.intermedia	W	94	6	0.202**	0.725 (0.178)***	0.864 (0.111)***	0.223 (0.100)***	83.3***	
	C	72	2	0.004	0.022 (0.019)	0.043 (0.037)	0.0003 (0.0003)	4.3***	
C.subternata	W	93	6	0.069**	0.332 (0.344)*	0.493 (0.375)*	0.049 (0.045)*	60.1***	
	C	72	3	0.013	0.026 (0.030)	0.050 (0.057)	0.0005 (0.0006)	1.7***	
C.longifolia	W	94	2	0.008	0.050 (0.055)	0.093 (0.102)	0.001 (0.002)	16.3	
	C	71	1	0	0 (0)	0 (0)	0 (0)	0	
Total	W	281	11	0.169**	0.645 (0.292)***	0.756 (0.298)***	0.154 (0.097)***	85.4***	
	C	215	3	0.074	0.262 (0.213)	0.394 (0.245)	0.0334 (0.028)	28.6**	
Notes.

* p < 0.05.

** p < 0.01.

*** p < 0.005.

Significance was determined from either Mantel test, Student t-test, or Wilcoxon rank sum test. Genetic variation represents variation between populations determined from AMOVA.

Discussion

This study set out to explore haplotype diversity patterns in wild and cultivated populations of Honeybush, an endemic neo-crop from the Cape Floristic Region (CFR) of South Africa. Cultivated populations appear to have originated from a small number of founding populations and/or individuals and represent a genetic bottleneck. Thus, cultivated populations tend to lack the genetic diversity and phylogeographic structuring present in wild populations and may represent a genetic threat to wild populations if gene-flow occurs.

Origin of cultivated genetic diversity

Despite initial Cyclopia breeding material originating from multiple wild populations (Joubert et al., 2011, Fig. 1), screening for individuals with commercially desirable traits has likely removed much of the haplotype richness from commercial breeding stock. The transition to Honeybush cultivation therefore represents a genetic bottleneck—under representing rare haplotypes and homogenizing the cultivated genepool.

Based on the NJ clustering of populations (Fig. 3), cultivated C. intermedia have likely originated from populations located in the Langkloof (LK); C. subternata populations from the Tsitsikamma and/or Outeniqua mountains (these wild populations share the same common haplotype found in cultivated populations); while C. longifolia originates from its only known wild source, the Van Stadens River system. These findings are consistent with those from recent microsatellite analysis of C. subternata (Niemandt et al., 2018). They compared wild populations from the Tsitsikamma and Outeniqua mountains to the Agricultural Resource Council’s (ARC) commercial genebank (an important source of commercially traded Honeybush seed), revealing no genetic differentiation between the two wild populations sampled and the genebank accessions. The lack of haplotype diversity detected in cultivated populations is therefore unlikely to be a byproduct of failing to detect variation in the slow evolving chloroplast genome (Schaal et al., 1998), as greater divergence was detected among populations originating from the Tsitsikamma or Outeniqua mountains using cpDNA screening via HRM than in the microsatellite based study.

The history of the movement of cultivated seed remains speculative and conversations with cultivators during sampling revealed that seed is largely sourced from existing farmer networks in their respective areas (with the initial origin of seed unknown). However, C. subternata cultivated in Harkerville (H) was confirmed to have been established from commercial seed and did not differ in dominant haplotypes from other cultivated populations. Furthermore, Harkerville shared a rare haplotype with cultivated material from Uniondale (U)—where putative hybrids were detected (NC Galuszynski, pers. obs., 2018, leaf material from these individuals was collected and is stored at the Nelson Mandela University in Port Elizabeth, South Africa). This rare haplotype was only detected in wild C. intermedia populations located in the Swartberg Mountains (SWB) and may be evidence of possible chloroplast capture (Hansen, Siegismund & Jørgensen, 2003) resulting from interspecific crosses taking place during the initial breeding trials (Joubert et al., 2011) or under field cultivation. The reasoning behind this argument is two fold. First, the chloroplast regions screened exhibit phylogeographic structuring in both species (Galuszynski & Potts, 2020a; Galuszynski & Potts, 2020c) and it is therefore unlikely that a wild C. subternata population will support this rare haplotype outside of the Swartberg mountains (where C. subternata does not naturally occur). Secondly, C. intermedia and C. subternata have been found to successfully produce hybrid offspring with other members of the genus under experimental conditions, and C. intermedia material from the Swartberg was included in the initial Honeybush cultivation trials (Joubert et al., 2011, Fig. 1). However, interspecific hybridization should be investigated through additional molecular work targeting the nuclear genome, which is subject to recombination and provides more insight into introgression history.

Potential impacts of cultivated genetic material on wild populations

Higher levels of genetic diversity were detected in wild Cyclopia populations than cultivated populations, with multiple cases of near complete haplotype turnover among wild populations (Table 1)—cultivated populations exhibited nearly no differentiation (Table 3). This level of haplotype turnover and genetic structuring of cpDNA in the wild was expected and has been described in detail for C. intermedia (Galuszynski & Potts, 2020a) and C. subternata (Galuszynski & Potts, 2020c). This suggests that there may be a tendency for cultivated populations to fail to account for natural phylogeographic patterns in regions where genetic structuring of plant populations occurs, including regions that may play a significant role in the discovery of neo-crops (e.g., Mesoamerica, Ornelas et al., 2013; South America, Turchetto-Zolet et al., 2013; Australia, Byrne, 2008; the Mediterranean basin, Feliner, 2014; and the Cape of South Africa, Galuszynski & Potts, 2020b; Tolley et al., 2014).

Genetic diversity differed between species (Table 3), suggesting different demographic histories among these closely related taxa (Ellegren & Galtier, 2016). Notably, rare and locally endemic taxa are predicted to have low levels of genetic variation due to their restricted distributions and small population sizes (Ellstrand & Elam, 1993; Gitzendanner & Soltis, 2000; Segarra-Moragues, Torres-Díaz & Ojeda, 2012), evident in wild C. Longifolia populations. The redistribution of genetic lineages via neo-crop cultivation needs to account for differences in demographic histories. Local endemics, for instance, may be at greater genetic risk from cultivated variants due to naturally low levels of genetic variation being more susceptible to genetic pollution by foreign lineages (Levin, Francisco-Ortega & Jansen, 1996; Wolf, Takebayashi & Rieseberg, 2001). Commercial production of narrow endemics may therefore require periodic supplementing of cultivated stands with locally sourced seed material in order to promote the preservation of rare haplotypes.

Genetic pollution can only occur if genetic material is able to escape into the wild. Seed dispersal is limited to a few meters in Cyclopia, and one would expect the chances of seed escape to be low. However, unmonitored spillover of cultivated seed into adjacent natural habitat does occur (NC Galuszynski, pers. obs., 2017) in addition to rare cases of cultivated plants intentionally established in natural vegetation (NC Galuszynski, pers. obs., 2017; G McGregor, pers. com., 2016; S Nortje, pers. com., 2019). The chloroplast genome is maternally inherited with no recombination in the majority of angiosperms (Mogensen, 1996) and is, therefore, subject to directional selective sweeps. Thus, the introduction of vigorous maternal lineages from commercial seed lots could disrupt local haplotypes diversity in wild crop plants. While the impacts of foreign haplotypes establishing in wild Honeybush populations is unknown, the introduction of foreign Spartina alterniflora (Poaceae) haplotypes from the United States to native Chinese populations resulted in a hybrid swarm that disrupted local cpDNA genome frequency and ecological processes due to vigorous growth displacing native plant species (Qiao et al., 2019).

Mass flowering of cultivated populations forms a powerful attractant to pollinators, increasing local pollinator density (Holzschuh et al., 2011; Westphal, Steffan-Dewenter & Tscharntke, 2003). This facilitates the spread of genetic material from cultivated populations into the wild via pollen flow, particularly in outcrossing species (Ellstrand, Prentice & Hancock, 1999), such as members of Cyclopia (Koen et al., 2021). By altering local allele frequencies through the introduction of large genetically depauperate commercial plantations, a landscape wide genetic bottleneck may result, promoting erosion of wild genetic diversity. The current state of neo-crop cultivation, relying on low genetic diversity breeding stock that is redistributed outside of its natural range (and possibly containing interspecific hybrid taxa as observed in the Honeybush populations in Uniondale, but requiring further research), represents a genetic threat that should be acknowledged and mitigated.

Until the genetic risks are better understood, formal guidelines should be developed to facilitate sustainable cultivation of neo-crops. In the case of Honeybush, an ecology-centric approach, as the one outlined by Potts (2017), may be desired due to the conservation value of many Cyclopia species; including the commercially important taxa: C. longifolia (critically endangered), C. genistoides (near threatened), C. maculata (near threatened), C. plicata (endangered), and C. sessiliflora (near threatened). However, it is unlikely that all neo-crops will exhibit the levels of phylogeographic structuring, genetic diversity and, conservation requirements of Cyclopia, and more work is required to define the extent to which wild genetic resources are at risk during neo-crop cultivation practices, particularly in species rich landscapes such as the Cape of South Africa.

Supplemental Information

Supplemental Information 1 Laboratory methods

Click here for additional data file.

Supplemental Information 2 Custom R script

Working example and R script used to assign haplotype identities to assecions based on HRM clustering results.

Click here for additional data file.

Supplemental Information 3 ndh A sequence data

Click here for additional data file.

Supplemental Information 4 ATPI-ATPH haplotype sequence

Click here for additional data file.

Additional Information and Declarations

Competing Interests

Author Contributions

Field Study Permissions

DNA Deposition

Data Availability

The authors declare there are no competing interests.

Nicholas C. Galuszynski conceived and designed the experiments, performed the experiments, analyzed the data, prepared figures and/or tables, authored or reviewed drafts of the paper, and approved the final draft.

The following information was supplied relating to field study approvals (i.e., approving body and any reference numbers):

Cape Nature (Permit number: CN35-28-4367), the Eastern Cape Department of Economic Development, Environmental Affairs and Tourism (Permit numbers: CRO 84/ 16CR, CRO 85/ 16CR), and the Eastern Cape Parks and Tourism Agency (Permit number: RA_0185).

The following information was supplied regarding the deposition of DNA sequences:

All novel DNA sequences are available at GenBank: MN930746–MN930919 and MW316480–MW316534.

The following information was supplied regarding data availability:

The HRM clustering to haplotype assignment for all samples included in the manuscript is available at figshare: Galuszynski, Nicholas (2020): HRM culstering to haplotype assignment:wild vs cultivated Honeybush. figshare. Dataset. https://doi.org/10.6084/m9.figshare.11370477.v1

The custom R script used to assign haplotype identity to samples based on HRM clustering is available as supplementary material and at figshare: Galuszynski, Nicholas (2020): R script used to assign haplotype identity to samples based on HRM clustering. figshare. Software. https://doi.org/10.6084/m9.figshare.12624620.v1.

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
