# Peer review of "The transition to agricultural cultivation of neo-crops may fail to account for wild genetic diversity patterns: insights from the Cape Floristic Region"

_PeerJ, doi:10.7717/peerj.11462_

## Round 0.1 · original submission · Minor Revisions

You will find below a number of issues raised by the three reviewers, two of them coincided (and I agree) that changes are needed to better explain methods. In addition more detail in explaining hybridization and how material was collected for the study. is needed as well. Two of the reviewers included a file with their suggestions, please take them into account.

Reviewer 1 ·

Basic reporting

The study presents a comparison of genetic diversity between wild and cultivated accessions of three species of Cylcopia using chloroplast haplotype analysis. The manuscript is generally well written and data analysis seems to be adequate. References seem to be adequate, as well as the article structure and data presentation.
The study also highlights a very pressing topic, because the cultivation of wild species is an increasing occurring phenomenon but consequences for wild species are currently not well considered. There are some minor points which I would suggest to include in the manuscript and which I describe in the sections below

Experimental design

Research question is well defined and methods are sufficiently described.
There are some points regarding the method section where improvements can be suggested:
The author published two other papers in the context, one about C.i. and one about C. s., the latter introducing the method. The paper at hand should be clear to what extent samples and sequences had been already used in the other publications. This should be clearly broken down in the method section, How many samples included and how many where already published. If the all had been collected newly, this can be stated. Ideally the previous studies are already mentioned in the introduction as previous results.

The description of the study system:
It is unclear to what extent the species hybridize. The existence of haplotypes shared between species can have different causes. Cultivated C subterranus seem to share haplotypes with C. i. but not the non cultivated ones. In this context it is of interest how well the species are circumscribed morphological and how their phylogenetic position within the genus is. It might be useful to include this as part of the Introduction or the study system section in the methods and introduce: morphological circumscription, relative phylogeny of the species, and hybridization potential. This section should also contain the description of the commercial circumscription of honeybush. In line 96 it is said that the three most widely used species for commercialization are included in the study. Interesting questions are: how many species Honey bush has, How many in the cape region, How many of these are cultivated? And in the discussion, if there is the possibility that other species than the included ones influence the study. There are a number of Honeybish species mentioned in the end of the discussion that also seem to be of commercial interest. This should be clarified.
Line 98- 99 is a very fundamental assumption of the dataset. It should be elaborated more clearly. If not in the discussion, then in the method section.
101ff resprouting is an ecological characteristic of having vegetative dispersal possibility. The differences between the species might also influence outcrossing behavior, this is more important in the context at hand and should be clearly specified. An obligate resprouter, shows only vegetative propagation? Geneflow between populations should be of no concern then.
The general Method:
The cost effectiveness of the method is mentioned, however, sequencing is today not anymore very time consuming neither too expensive. Even if the method is an alternative for sequencing, the effort should be more clearly outlined. For example if specific instruments or software is needed, should be mentioned at least in a few sentences.
minor remarks:
148 contemporary sanger sequencing is very advanced over the original Sanger 1977 approach. The reference for fluorescence labelled dd nucleotides should be more fitting. However, this is a standard method for which a reference is not necessary.
187 to 189: this information should be part of the methods .

Validity of the findings

Sequences had been deposited in genebank. Conclusions are clear, however, some points are not considered which might be important in this context

The Results first paragraph: The description about haplotype distribution in the populations is difficult to follow.

In the discussion I suggest to include a paragraph about the significance of chloroplast diversity for the question of genetic introgression of cultivated material into wild populations. When the genetic exchange is via pollen, then Chloroplast will not be effected. Pollination of plantations with pollen from the surroundings will increase diversity but not at chloroplast loci. The peculiarities of cpDNA variation should be reflected.
The suggestion of an ecocentric approach could be more clearly reflected with the literature. There are also different goals to be considered, a breeding / selection aspect for the crop requiring a reduction of diversity and the conservation aspect requiring a high diversity. The latter resembles translocation procedures and not procedure for agricultural production. This should be reflected. Refreshing a crop regularly with diversity from wild accessions, might have a conservation implication with the very rare species.

In line 215 genetically distinct, Please be more precise from what.

·

Basic reporting

The manuscript was well written and easy to read. A few minor comments were included and should be adressed but does not distract from the merit of the article.

Experimental design

I found the sampling design well structured and supported by a well planned and executed experimental design in the form of a validated HRM technique. Appropriate genetic differentiation analyses were conducted.

Validity of the findings

I found the findings important in the context of this genus and it add to be growing body of knowledge for this important indigenous genus. I appreciated the recommendations that were included.

Additional comments

An important study and I applaud the author for taking an "alternative" perhaps not always appreciated route in terms of HRM and supported sequencing to address the research question.

Reviewer 3 ·

Basic reporting

This is a timely paper, and highly relevant to the honeybush and other Fynbos sectors. It's significance for other sectors similar to that of honeybush (e.g., cutflowers (Protea spp.), buch (Agathosma spp.) and rooibos (Aspalathus linearis) is not really mentioned, though. It may be reasonable to assume that this was not an oversight: the authors may have wanted to limit themselves to the scope of their data, which is specific to honeybush.

Language use was good, but some parts need a bit of editing for clarity and consistency. Some sections (e.g. the first and last parts of the "methods" section) are very well written; consistency would help to even out the tone of the entire document.

The literature references and field background are comprehensive, and include the latest work relevant to the paper.

The structure of the paper, and the content of figures and tables conform with convention. Captions, however, need some work here and there. Caption titles should make contextual sense as stand-alone items in the paper. In two of the captions there is no reference to the genus name, and in one, there is no reference to the species or genus reflected in the content of the table. This should be corrected.

Results and scope are relevant to the objectives set out.

Conceptually and methdologically, this is a sound paper. It raises good points that are highly relevant to the honeybush sector, and does so based on good reasoning about ecological concepts. Responses to the comments, as well as correction of a few grammatical and formatting errors (also found in the bibliography) will see this paper fit for publication.

Experimental design

Site selection and sample sizes can be difficult in production areas - there are examples of studies where these are limiting factors (e.g. Chimpango and Hattas, 2017; Potsma et al. (2014).

As mentioned above, the research is timely and highly relevant. Of the two objectives, the latter didn't seem to do justice to the significance of the study. Please see the annotated paper for details.

It isn't clear whether the attempt to avoid genetic material from the same seed lots during site selection was successful, but given that seed sources are mostly unknown (mentioned in the discussion), it probably isn't possible to be absolutely sure. It might be useful just to add a qualifying sentence or two, making this explicit in the methods section.

Methods are well described. Minor indications of information that is unclear are highlighted in the text. Statistical reporting conventions are adhered to, and there is no indication of anything but rigorous analysis. Although there is no indication of unethical research practices, it would be good for authors to say something about and the permissions obtained to access research sites. Presumably, sites were located on private or state-owned land, in which case, such permissions would have been necessary.

The authors should include the time and timing of sampling efforts. Of course, there may be little implication for a genetic study like this one, but it does add to methodological and reporting rigour.

Validity of the findings

Findings are novel, and build on a previous study that first raised the usually obtuse fact of gene distribution in production systems of these endemic species. The study should stimulate research in related sectors, like cutflowers and rooibos: together with honeybush, these are world exports from the CFR that are earmarked for expansion under the Green Economy for social and economic development, with easy departure from the ecological principles that should govern production and management. This research sheds light on one of the least considered amongst the ecological parameters for sustainable biomass production.

The building of reference material from disparate wild populations was appropriate, and by all indications, was well executed.

Conclusions are well stated, fairly well made, and conclusions aligned well with results and objectives.

Additional comments

This paper is a timely and necessary contribution to research on honeybush, and holds promise for similar lines of investigation in related Fynbos sectors. Despite a growing body of research on these commercial genera, little has been done to advance work on their genetics. Historically, research has been strongly industry-driven, with a focus on traits that might advance production and marketing of standardised biotypes. It is only in recent times that phytochemical profiling, and now, genetic research, is raising the profile of plants in the wild.

Your paper raises the need for a more definitive shift to ecologically-centred management and production for honeybush. The paper is timely - if repackaged for the appropriate audience, this study could feed into local government efforts to shape policy around production and management.

Implications for other Fynbos-based sectors are not highlighted in the paper, but is not to be underestimated. It would be good to follow the lines of investigation into hoenybush, but also to think laterally across other genera and sectors that might benefit from these important ecological insights.

Annotated reviews are not available for download in order to protect the identity of reviewers who chose to remain anonymous.

---

## Round 0.2 · Minor Revisions

Thank you for considering the issues raised by the reviewers, the Revision 1 version is much improved. My only concern is the quality of Figure 1, it is not possible to understand distribution and sampling in a white and black image. Could you please consider this suggestion?

---

## Round 0.3 · Minor Revisions

Thank you for considering all issues raised by the three reviewers. When I was reading the revised manuscript, I realized that the use of the Honeybush is not indicated in the Abstract. I suggest including its use in this section, this is my only suggestion.

---

## Round 0.4 · accepted · Accept

The figure improved and now It is possible to identify where the populations were collected. Thank you also for including the use of the studied species in Abstract, it reads much better.